# Breaking the fundamental scattering limit with gain metasurfaces

Chao Qian ®[1,2,3] ✉, Yi Yang ®[4,5], Yifei Hua ®[1,2,3], Chan Wang[1,2,3], Xiao Lin ®[1,2,3] ✉, Tong Cai[1,2,3], Dexin Ye ®[1], Erping Li ®[1,2,3], Ido Kaminer[6] & Hongsheng Chen ®[1,2,3] ✉

A long-held tenet in physics asserts that particles interacting with light suffer from a fundamental limit to their scattering cross section, referred to as the single-channel scattering limit. This notion, appearing in all one, two, and three dimensions, severely limits the interaction strength between all types of passive resonators and photonic environments and thus constrains a plethora of applications in bioimaging, sensing, and photovoltaics. Here, we propose a route to overcome this limit by exploiting gain media. We show that when an excited resonance is critically coupled to the desired scattering channel, an arbitrarily large scattering cross section can be achieved in principle. From a transient analysis, we explain the formation and relaxation of this phenomenon and compare it with the degeneracy-induced multi-channel superscattering, whose temporal behaviors have been usually overlooked. To experimentally test our predictions, we design a two-dimensional resonator encircled by gain metasurfaces incorporating nega- tive- resistance components and demonstrate that the scattering cross sec- tion exceeds the single- channel limit by more than 40-fold. Our findings verify the possibility of stronger scattering beyond the fundamental scat- tering limit and herald a novel class of light-matter interactions enabled by gain metasurfaces.

Electromagnetic (EM) scattering describes a fundamental process where an obstacle continuously removes energy from the incident wave and then re-radiates energy in all directions. The balance between in and out-coupling of photons ultimately determines the resultant scattering and absorption cross-sections[1]. These cross-sections are bounded from above. Considering a two-dimensional (2D) particle, its total scattering cross section is the scattering sum from all angular momentum channels:

$$C_{\text{sct}} = \frac{2\lambda}{n\pi} \sum_{m=-\infty}^{\infty} |S_m|^2 \qquad (1)$$

where $S_m$ is the scattering coefficient of the $m$ th angular momentum channel, $\lambda$ is the wavelength of light in free space, and $n$ is the refractive

[1]ZJU-UIUC Institute, Interdisciplinary Center for Quantum Information, State Key Laboratory of Modern Optical Instrumentation, Zhejiang University, Hang-zhou 310027, China. [2]ZJU-Hangzhou Global Science and Technology Innovation Center, Key Lab of Advanced Micro/Nano Electronic Devices & Smart Systems of Zhejiang, Zhejiang University, Hangzhou 310027, China. [3]Jinhua Institute of Zhejiang University, Zhejiang University, Jinhua 321099, China. [4]Department of Physics and Research Laboratory of Electronics, Massachusetts Institute of Technology, Cambridge, MA 02139, USA. [5]Department of Physics, University of Hong Kong, Hong Kong, China. [6]Department of Electrical and Computer Engineering, Technion-Israel Institute of Technology, Haifa 32000, Israel. ✉e-mail: chaoq@intl.zju.edu.cn; xiaolinzju@zju.edu.cn; hansomchen@zju.edu.cn

**Fig. 1 | Breakdown of the single-channel scattering limit by exploiting gain materials. a** Scattering cross-section from an individual channel $m$ as a function of $\gamma_{\mathrm{loss}}^m / \gamma_{\mathrm{leak}}^m$ based on the temporal coupled theory, where $\gamma_{\mathrm{loss}}^m$ and $\gamma_{\mathrm{leak}}^m$ are the intrinsic loss rate and external leakage rate, respectively. Conventional scattering-enhanced works mostly focus on the right shaded green region ($\gamma_{\mathrm{loss}}^m > 0$, lossy system), which is fundamentally bounded by the single-channel scattering limit, i.e., $|S_m| < 1$. More crucially, the desired performances are often hindered by the lack of

low-loss materials and delicate designs in practice. Here, we challenge this established single-channel scattering limit by exploiting gain materials ($\gamma_{\mathrm{loss}}^m < 0$). **b** Schematic illustration of the gain-assisted scattering enhancement approach. We term this scheme a single-channel superscatterer because it overcomes the single-channel scattering limit. **c** Schematic illustration of the multi-channel superscatterer. This scheme overlaps the scattering contribution of multiple channels via degenerate resonance, which has been widely studied[3–7].

index of the surrounding environment[2] (Supplementary Note 1). In a passive system, one can rigorously prove that the scattering cross section from each individual channel is bounded by $|S_m| \leq 1$, referred to as the single-channel scattering limit[3–6]. A similar limit also exists in three-dimensional (3D) situations[7,8].

Enhancing the scattering cross section is attractive to a wide range of applications, such as photovoltaics, lasers, sensing, and spectroscopy[9–12]. To this end, there have been intense research activities exploiting different methods over the past decades[3–8,13–17]. Based on the above formula, the existing methods can be generally categorized into two types (the right region of Fig. 1a). The first type is to engineer a superposition of scattering from as many channels as possible due to their orthogonality. A facile way to achieve this purpose is to magnify the physical size of a given object because more high-order dipole moments will take effect, for example, using transformation optics to equivalently construct a large scatterer[13]. However, experimentally achieving complementary metamaterials with both the desired anisotropy and inhomogeneity poses a great challenge. Distinct from this technique, a superscattering method boosts scattering by overlapping multiple scattering peaks with near-degenerate plasmonic modes[3–7]. This method only requires an isotropic medium, but it hardly reaches a scattering cross section greater than 10 times due to the unique construction of subwavelength-layered resonators[14]. The second type is to increase the pre-factor of Eq. (1), i.e., decrease the refractive index of the surrounding environment. For example, one viable way is to embed a resonator in near-zero index (NZI) materials, which can be constructed via photonic crystals and waveguides[14,15]. Evidently, this method relies on extreme photonic environments, while in essence, it does not improve the channel-normalized scattering cross section $|S_m|$ in Eq. (1).

According to the above analysis, all existing approaches remain bounded by the fundamental single-channel scattering limit, despite the total scattering enhancement they enable. Particularly, when realistic material losses are considered, the scattering-enhanced effects based on these routes could severely deteriorate or even disappear in practical implementations (see the example in Supplementary Fig. 4a). Therefore, it would be enticing, both fundamentally and application-wise, if there exists a third scattering-enhanced route that can overcome the single-channel scattering limit (the left region of Fig. 1a).

In this work, we show how to break this "cage" by exploiting gain media. According to the temporal coupled-mode theory[18–20], the scattering coefficient of the $m$th angular momentum channel can be expressed as $|S_m| = \left| \frac{\gamma_{\mathrm{leak}}^m}{i(\omega_0 - \omega) + \gamma_{\mathrm{loss}}^m + \gamma_{\mathrm{leak}}^m} \right|$, where $\omega_0$ is the resonant frequency (Supplementary Note 2). $\gamma_{\mathrm{loss}}^m$ and $\gamma_{\mathrm{leak}}^m$ are the intrinsic loss rate and external leakage rate, respectively, which characterize the energy dissipation mode via generation of Joule heat and coupling to the far field, respectively. Here, we find that for $\gamma_{\mathrm{loss}}^m < 0$ in a gain system, the scattering coefficient could increase remarkably. In principle, an infinitely large, resonant (i.e., $\omega = \omega_0$) scattering cross section could be anticipated at $\gamma_{\mathrm{loss}}^m = -\gamma_{\mathrm{leak}}^m$. One may naively conceive that the single-channel scattering limit is bound to be overcome by introducing gain, even proportional to the gain. However, gain is a *necessary but insufficient* condition to overcome the single-channel scattering limit, as exemplified in Supplementary Fig. 4b. This occurs because the above theory only considers a set of essential components that are analyzed based on only very general principles, and the unknown parameters depend on the specific geometry[18].

## Results

### Single-channel superscatterer

For conceptual clarity, we term this gain-assisted scattering-enhanced method as single-channel superscattering because the total scattering can be extremely enhanced from only one angular momentum channel (the upper inset of Fig. 1a). In particular, we denote the previous degeneracy-induced superscattering method as multi-channel superscattering (the lower inset of Fig. 1a)[3–7]. In contrast to multi-channel superscattering, as well as other methods in the right region of Fig. 1a, single-channel superscattering does not rely on multiple plasmonic resonances or extreme environments and thus mitigates the high requirements on the geometrical complexity, material loss, and refractive indices of the surrounding environment. Instead, we should compensate for leakage via a properly engineered gain profile both temporally and spatially. If the single-channel scattering limit is overcome, the total scattering cross section should be at least two times the single-channel scattering limit, i.e., $C_{\mathrm{sct}} > 2 \times \frac{2\lambda}{n\pi}$, because of the consistent degeneracy of the $m$th and $-m$th angular momentum channels (i.e., $S_m = S_{-m}$).

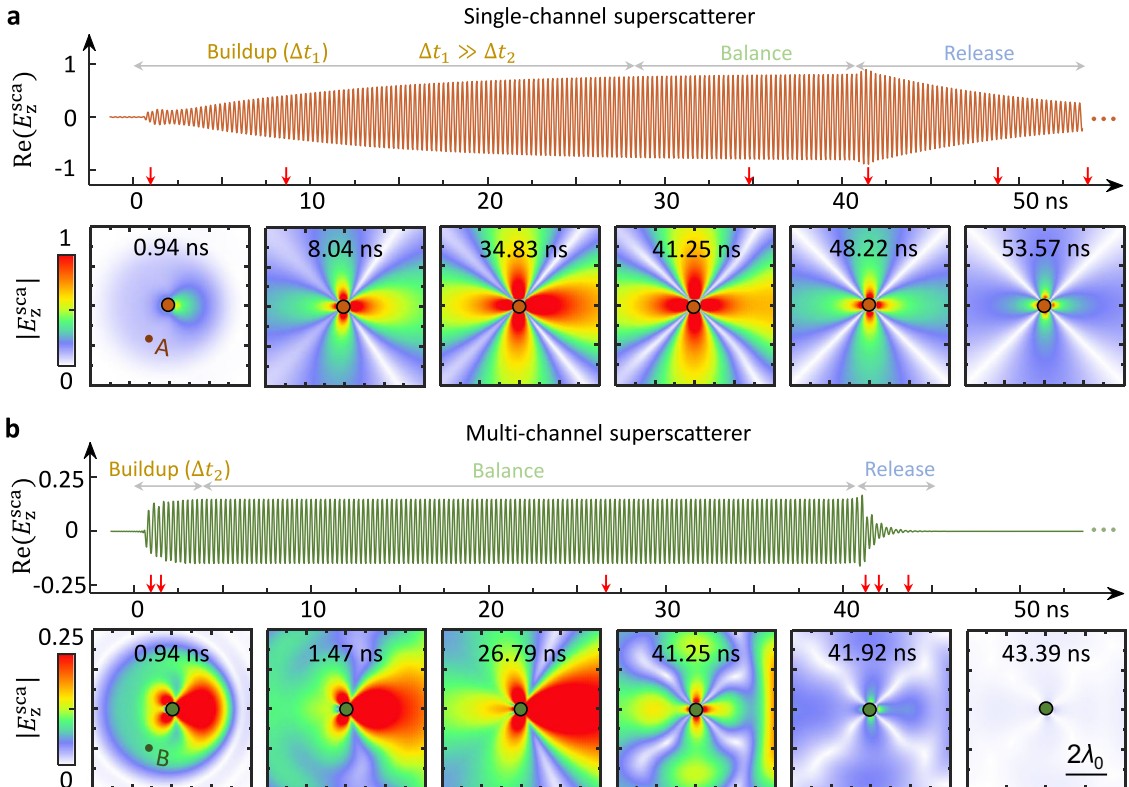

**Fig. 2 | Transient response of the single-channel superscatterer in the time domain.** Here, we consider the incidence of a time-modulated plane wave from left to right, which is modulated by a time window from 0 ns to 40.18 ns, as shown in Supplementary Fig. 8. The modulated frequency is 3.733 GHz (with a period of 0.27 ns), in accordance with the working frequency of the two superscatterers. **a** Time evolution of the scattered field $E_z^{sca}$ at point A (upper panel) and a region (lower panel) of the single-channel superscatterer. The whole temporal process is

summarized into three stages, namely, buildup, balance, and release, as schematically shown in the figure. **b** Time evolution of the scattered field $E_z^{sca}$ at point B and a region of the multi-channel superscatterer. As a key feature, the single-channel superscatterer requires a much longer time to accumulate energy than the multi-channel superscatterer, i.e., $\Delta t_1 \gg \Delta t_2$. For further comparison, the transient response of a homogeneous dielectric rod with much smaller scattering is shown in Supplementary Fig. 9.

We consider a 2D single-channel superscatterer at microwaves for demonstration. An ultrathin metasurface encircling a subwavelength Teflon rod (with a relative permittivity of 2.1) offers a complex surface impedance $Z_s$ or surface conductivity $\sigma_s$ ($=1/Z_s$), which can be flexibly designed via geometrical means and the addition of gain components[21]. The real part of $Z_s$ determines the lossy or gain property, and its imaginary part determines the capacitive or inductive reactance. We leverage the Mie theory[2] and annealing optimization[22] to determine the structural parameters of our scatterer. Specifically, we choose a subwavelength Teflon rod diameter of 28 mm (0.35 times the working wavelength), working under transverse electric (TE, with the electric field along the cylinder symmetry axis) waves at $f_0 = 3.733$ GHz ($\lambda_0 = 80.4$ mm).

**Transient response**

The transient response of the single-channel superscatterer is crucial to better understanding its behaviors. As shown in Supplementary Fig. 8, we consider a temporal signal $g(t)$, with its associated Fourier transform of $G(\omega)$, impinging on the scatterer. According to the Fourier theory, in regard to a shifted signal $f(x, y, t) = g(t - x/c)$, its Fourier transform becomes $G(\omega) e^{ikx}$, where $c$ is the speed of light in free space and $k = \omega/c$ is the wavenumber. Notably, component $e^{ikx}$ exactly corresponds to a monochromatic plane wave propagating along the $x$ direction in $k$ space, i.e., $E_z^{inc}(x,y,\omega) = e^{ikx}$. The monochromatic plane wave passing through the scatterer thus generates scattered waves, denoted as $E_z^{sca}(x,y,\omega)$. For our system (the gain/loss is epitomized in the real part of $Z_s$), the time-evolution scattered wave can be directly

obtained as[23]:

$$E_z^{sca}(x,y,t) = \frac{1}{2\pi} \int_{-\infty}^{\infty} G(\omega) E_z^{sca}(x,y,\omega) e^{-i\omega t} d\omega \qquad (2)$$

where $E_z^{sca}(x,y,\omega)$ can be rigorously calculated via the Mie theory (Supplementary Note 1).

We consider a modulated window signal passing through both the single-channel superscatterer and a multi-channel superscatterer with the same radius and working frequency for comparison (Supplementary Note 4). Under the lossless assumption, the multi-channel superscatterer only attains a -1/10 scattering strength of the single-channel superscatterer, as shown in Supplementary Fig. 3. Specifically, $g(t) = e^{-i2\pi f_0 t + i\pi/2}(u(t) - u(t - t_{cut}))$, where $u(t)$ is a step function, $f_0$ is the working frequency, and $t_{cut} = 40.18$ ns. The integral range in Eq. (2) is set from 0 to 7 GHz with a step of 0.1 GHz. Figure 2 shows the time evolution of the scattered field of the single-channel and multi-channel superscatterers. When the incident wave moves close to the single-channel superscatterer (Fig. 2a), localized surface waves (spoof surface plasmons) are excited and gradually amplified via an external gain supply. After a period of $\Delta t_1 = 28.5$ ns, the spatial field stabilizes (the saturated gain effect is not considered in the simulation). In stark contrast, the multi-channel superscatterer requires a short buildup time ($\Delta t_2 = 4.1$ ns) (Fig. 2b), as well as a homogeneous dielectric rod ($\Delta t_3 = 0.8$ ns), as shown in Supplementary Fig. 9.

At $t = t_{cut}$, when the incident wave cuts off abruptly, we find that the field scattered by our single-channel superscatterer is deformed and very slowly decays thereafter. We summarize the entire process

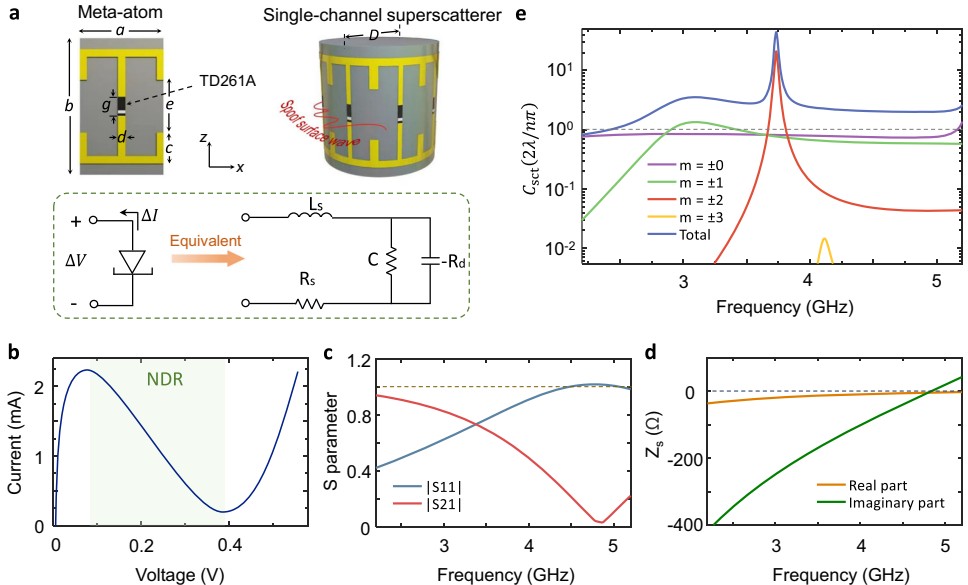

**Fig. 3 | Experimental design of the gain metasurfaces and the single-channel superscatterer in the microwave regime. a** Schematic of the designed single-channel superscatterer and the equivalent circuit model of deployed TD 261 A cells. The single-channel superscatterer comprises a subwavelength Teflon rod encircled by a flexible layer of gain metasurfaces, which contains eight meta-atoms. Inside each meta-atom, a tunnel diode (TD261A) is welded to provide a negative differential resistance (NDR, namely, gain); see its equivalent circuit model in the figure. The dimensions of each meta-atom are $a = 11$, $b = 18$, $e = 7$, $c = 4$, $d = 1$, and $D = 28$, all in millimeters. **b** Current–voltage curve of the tunnel diode. The colored region corresponds to the occurrence of NDR. **c** $S$ parameter of the planar gain metasurfaces. **d** Equivalent surface impedance $Z_s$ of the planar gain metasurfaces. As a key feature of the gain metasurfaces, we obtain $|S_{11}|^2 + |S_{21}|^2 > 1$ and $Re(Z_s) < 0$ at all frequencies in (c-d). **e** Analytical total scattering cross section and the scattering contribution of each individual channel. Due to the consistent degeneracy of the $m$th and $-m$th channels, $S_m = S_{-m}$. At 3.733 GHz, the scattering cross section from channel $m = \pm 2$ reaches 21 times the single-channel scattering limit.

and its dynamics into three stages: buildup (formation), balance, and release (relaxation) of energy. Full animation can be found in Supplementary Movie 1. We conclude that steady-state cross-sections also affect the transient behavior of scatterers—the larger the total scattering cross-section, the longer the time required to accumulate sufficient energy to reach a stable state. In a similar vein, the accumulated energy is stored for a longer period after the incident wave is cut off. For the gain system, the radiation leakage is compensated by the gain metasurfaces in the release process that artificially increases the mode lifetime.

## Gain metasurfaces

We next proceed to the design of the gain medium. As discussed in Supplementary Note 3, the key design principle is to obtain a uniform spatial distribution of negative conductance. In the microwave regime, this can be achieved by embedding negative-resistance components into metasurfaces, such as amplifiers and tunnel diodes (TDs)[24,25]. Due to the quantum mechanical effect, the TD can offer a negative differential resistance (NDR) controlled by an applied DC voltage. Here, we choose a germanium TD (General Electric's TD261A), whose current–voltage characteristics and equivalent circuits are shown in Fig. 3a, b, respectively. In the NDR region (from 0.08 V to 0.39 V), TD261A is effectively composed of a negative resistance $-R_d$ and parasitic components $R_p$, $L_p$, and $C_p$ due to the device package. At a bias voltage of 0.2 V, the TD261A parameters are $R_p = 7\,\Omega$, $L_p = 1.5$ nH, $C_p = 0.65$ pF and $-R_d = -250\,\Omega$. The periodicity of the subwavelength metasurface elements is 11 mm (-1/7$\lambda_0$). These elements are printed on a flexible substrate layer, with a relative permittivity of 3.4 and a thickness of only 0.05 mm. Figure 3c shows the simulated S parameters of the planar gain metasurfaces determined in commercially available CST Studio Suite full-wave simulation software. $|S_{11}|^2 + |S_{21}|^2 > 1$ within the entire plotted band, and the amplitude of the $S_{11}$ parameter is slightly higher than unity in the vicinity of 4.7 GHz. Based on the above, the surface impedance of the planar metasurface can be analytically

calculated with the equivalence, $Z_s = 2(1 - S_{21} - S_{11})/\eta(1 + S_{21} + S_{11})$, where $\eta$ is the wave impedance in free space[26]. As illustrated in Fig. 3d, the real part of the extracted surface impedance is always negative, which indicates that the metasurfaces operate in gain mode. The increase/decrease of $Re(Z_s)$ (gain component) will affect the total scattering cross section; see Supplementary Fig. 5. This lays a physical foundation to realize single-channel superscattering in tandem with further structural optimizations.

A metasurface guiding layer comprising eight metasurface elements along the $\phi$ direction conforms to the Teflon rod. It should be noted that the actually curved surface impedance may not be exactly the same as that extracted from the planar configuration. However, since the unit cells in our design are deep subwavelengths and there is no curvature along the polarization direction, the above analytical expression may still hold[27]. Figure 3e shows the analytical scattering cross-section of the designed single-channel superscatterer. Specifically, the total scattering cross section is up to 45 times the single-channel scattering limit at 3.733 GHz and exceeds previous scattering enhancements by one order of magnitude[3–7]. The total scattering cross section mainly originates from the $m = \pm 2$ channels (quadrupole mode), leading to a high-quality factor Q ($=f_0/\Delta f$) of approximately 113, where $f_0$ is the peak frequency and $\Delta f$ is the full frequency width at half maximum[6].

## Experimental measurement

A photograph of the fabricated structure is shown in Fig. 4a. In the experiment (Fig. 4b and S10), the fabricated sample is placed inside a parallel plate waveguide that only supports the fundamental TM$_0$ (transverse magnetic) mode at $f_0 = 3.733$ GHz. A horn antenna placed far from the waveguide is employed to generate a TE-polarized plane wave inside the waveguide. Before the experiment, we measured the current–voltage characteristics of the eight TDs (Supplementary Fig. 10) to verify their NDR properties. In the experiment, to guarantee that the gain metasurfaces operate in the linear amplification regime

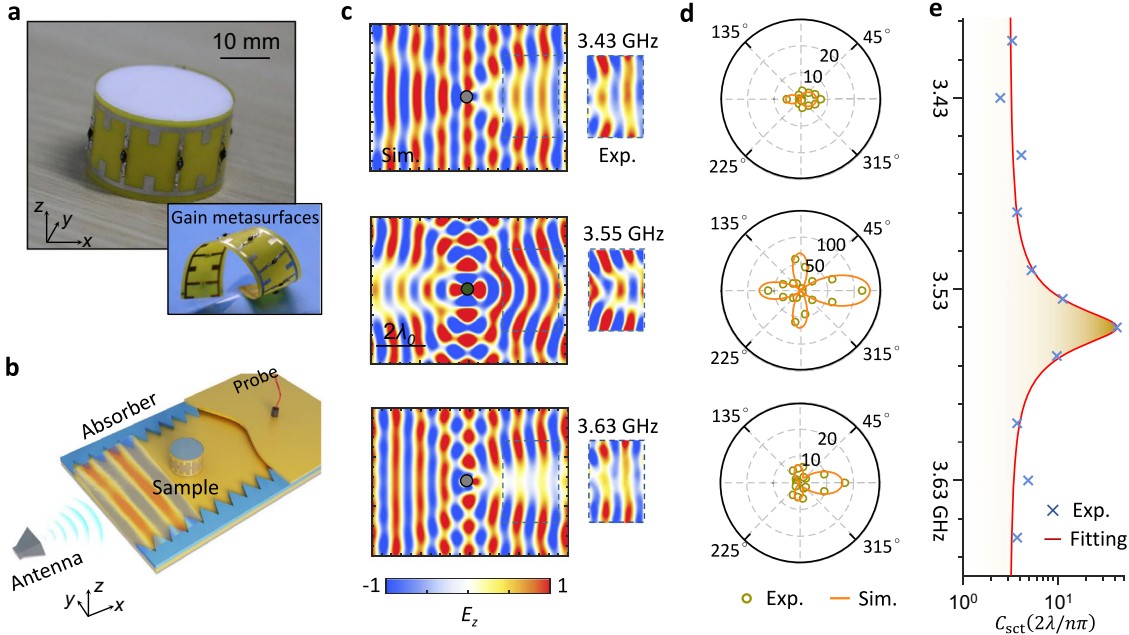

**Fig. 4 | Observation of the single-channel superscatterer in the frequency domain. a** Photograph of the fabricated single-channel superscatterer and gain metasurfaces. **b** A cutaway view of the experimental setup. To mimic a 2D case, the designed sample is located at the center of the two planar metallic waveguides separated by absorption layers with a distance of $h = 18$ mm, which is the same as the height of the single-channel superscatterer. Within the frequency range of interest, the designed waveguide only supports the fundamental $TM_0$ mode. A detailed description of the measurement is provided in Supplementary Note 6. **c** Near-field simulations and measurements of the total field in a region close to the single-channel superscatterer. To achieve the required gain, tunnel diodes inside each meta-atom are applied with a voltage of 0.2 V. **d** Measured scattering width at three frequencies. **e** Measured total scattering cross section by integrating the scattering width around one circle. For comparison, the related theoretical results are also shown in **c**–**e**, which attain a good match with the experimental results.

(no self-resonance), the input power is set to be lower than −45 dBm[25]. Through near-field scanning microwave microscopy, the electric field within a rectangular region of $272 \times 182$ mm² was measured. For reference, the analytical field distributions are also provided in Fig. 4c. A good agreement was achieved between the experimental results and simulations, despite a slight frequency shift (-0.15 GHz). This may be caused by the parasitic reactance of the TDs. Moreover, we should note another fact: the environmental noise is very low. Although the gain metasurfaces may somehow amplify noise (spontaneous emission), they yield a near-zero contribution to the observed scattering enhancement induced by external illumination[25].

To quantitatively characterize the far-field radiation pattern of the gain scatterer, we further measured the radar cross section/scattering width, i.e., the scattering per azimuthal angle, $C_{rcs}(\phi) = 2\pi\rho_0 |E_s|^2/|E_i|^2$, where $\phi$ is the angle between the wave vector of the scattered wave and $\hat{x}$ coordinate, $E_i$ and $E_s$ are the incident and scattered fields at $\rho_0$, respectively, and $\rho_0$ is the radial distance from the position of the scatterer[5]. In our experiment, $\rho_0 = 40$ cm is adopted to ensure that the measurement is carried out in the far field. As shown in Fig. 4d, the measured far-field pattern is generally consistent with the theoretical prediction, where the external gain greatly reshapes the radiation pattern. Based on these angle-resolved cross-sections, the total scattering cross-section can be easily obtained via $C_{sct} = \frac{1}{2\pi}\int_0^{2\pi} C_{rcs} d\phi$. Figure 4e shows the total scattering cross section at discrete frequencies, which is further fitted by the temporal coupled-mode theory (Lorentzian fitting curve) with the minimum least-squared error. The experimental result indicates that the total scattering cross section is >40 times the single-channel scattering limit at 3.55 GHz. To our knowledge, this scattering enhancement is higher than the previous achievements[3–7].

## Discussion
In conclusion, we introduced a novel scattering-enhanced method that overcomes the long-held single-channel scattering limit and thus generalizes the prevailing paradigms for scattering systems. Similar configurations may be potentially explored to maximize the absorption cross section by engineering the coupling state of resonators[28] and thereby resolve a subject of controversy persisting longer than half a century—total internal reflection in a gain medium[29]. The peculiar dynamics of superscattering build on a key example in illuminating puzzles in other exotic scattering phenomena, such as transformation optics-based effects[30] and free-electron radiation[23,31]. Other than scattering enhancement, the potential and versatility of NDR-enabled gain metasurfaces may proliferate across other fields of active metamaterials[32], nano-lasers[33], and non-Hermitian invisibility[34]. For example, we envisage a small-scale drone disguised as a large-scale aircraft to deceive enemy detection systems because the gain-assisted device can create a much larger scattering field than it's physical size[35–38]. The application of other gain materials could help extend the concept to higher frequencies, such as terahertz NDR diodes, optical pulse-pumped organic dye molecules, and quantum dots[39].

## Methods
### Analytical scattering calculation and optimization strategy
The scattering cross-section of the 2D layered rod is calculated based on the Mie scattering theory. By matching the boundary conditions, the field distribution in each region and scattering coefficient can be obtained. In the calculation, the ultrathin and flexible metasurfaces are modeled with a surface impedance $Z_s$. To calculate $Z_s$, the simulation program with an integrated circuit emphasis (SPICE) model and DC bias of TD261A are introduced. $Z_s$ is influenced by the geometries of the unit cell and gain components (input variables) and affects the scattering cross section (output). To this end, we transfer the unit cell into the commercial software package CST Microwave Studio for continuous automatic full-wave simulations via the MATLAB-CST co-simulation method. In conjunction with the simulated annealing algorithm, we iteratively optimize the structure until convergence is attained.

## Numerical simulation

The scattering parameters of the gain metasurfaces are simulated using CST Microwave studio. The gain metasurface comprises a copper patch and substrate layer with a relative permittivity of 3.4 and a thickness of 0.05 mm. A tunnel diode is incorporated into each metasurface element, whose equivalent circuits are embedded in the simulation. We applied the EM-circuit co-simulation to connect the metasurface and tunnel diode. The "FSS, Metamaterial-Unit Cell template" in the "Frequency domain solver" is chosen, with unit cell boundary conditions enabled in the x- and z-directions and open boundary conditions in the y-direction (Fig. 3a). Finally, an adaptively refined tetrahedral mesh was applied to the simulated structures with a number of about 35,000 cells. The maximum mesh size was set at ~4.8 mm, which is about one-eighteenth of a free-space wavelength at 3.5 GHz.

## Experimental measurement

The microwave experiment was performed in a planar waveguide to mimic a 2D situation. The measurement system mainly contains a transmitting horn antenna, a receiving probe, and a vector network analyser (Supplementary Note 9). In the near-field measurements, a small receiving probe, which can move programmatically, is employed to detect the spatial near-field distribution, including both its amplitude and phase, via transmission coefficient $S_{21}$. Before the experiment, we measured the near-field for the pure dielectric rod without gain metasurfaces to check the border effects (Supplementary Fig. 12). In the far-field measurements, we constructed an angle-resolved scattering measurement setup. Two sequential steps are measured: the incident field without the scatterer and the total field with the scatterer. With information on the incident and total fields, the scattered field can be obtained via subtraction, and the scattering width can then be calculated accordingly.

## Reporting summary

Further information on research design is available in the Nature Research Reporting Summary linked to this article.

## Data availability

Data presented in this publication is available on Figshare with the following identifier (https://doi.org/10.6084/m9.figshare.20179907.v1).

## Code availability

The codes used in the current study are available from the corresponding authors upon reasonable request.

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

## Acknowledgements

The work at Zhejiang University was sponsored by the National Natural Science Foundation of China (NNSFC) under Grant Nos. 62101485 (C.Q.), 61625502 (H.C.), 11961141010 (H.C.), 61975176 (H.C.), 62175212 (X.L.), 62071424 (E.L.), and 62027805 (E.L.), the Top-Notch Young Talents Program of China (H.C.), and the Fundamental Research Funds for the Central Universities under Grant No. 2021FZZX001-19 (H.C.), and Zhejiang University Global Partnership Fund (X.L.). Y.Y. thank the support from the start-up fund of the University of Hong Kong.

## Author contributions

C.Q., X.L., and H. C. conceived the idea; C.Q. conducted the numerical simulations and experiments with the help of Y.H. and C. W.; C.Q. wrote the manuscript with the input from X.L., Y.Y., and I.K. T.C., D.Y., and E.L. discussed the results; C.Q., X.L., and H. C. supervised the project.

## Competing interests

The authors declare no competing interests.

## Additional information

**Peer review information** *Nature Communications* thanks Patrice Genevet and Other anonymous Reviewer(s) to the peer review of this work. Peer review Reports are available.

