## [Peer Review File · Nature Communications]

REVIEWER COMMENTS

Reviewer #1 (Remarks to the Author):

In this manuscript, the authors proposed novel scattering enhanced method that overcomes the single-channel scattering limit by exploiting gain media. As a demonstration, the authors have design a two-dimensional resonator with tunnel diode in order to provide a negative differential resistance. Overall, the authors have presented an interesting concept to exploit gain metasurfaces.

The manuscript is well-written and structured. It shows studies on the dynamic of these media and it shows a progress in the scattering enhancement limit with respect previous achievements. However, some points must be addressed for publication in Nature Communications.

1-The authors claim that the total scattering cross section is more than 40 times the single-channel scattering at 3.55GHz, however no experimental point at the maximum enhancement is provided. Experimental point in this frequency must be provided in order to demonstrate the claimed enhancement.

2- An experimental measurement of the dielectric rod without gain metasurface could be very helpful to discard border effects in the electromagnetic set up measurements.

3-Taking into account that the experimental design and the results are based in the numerical simulation of CST studio commercial software, the numerical method must be more detailed (boundary conditions, mesh, how the metallic materials are considered, etc.)

Reviewer #2 (Remarks to the Author):

The manuscript entitled «Breaking the fundamental scattering limit with gain metasurfaces » by Chao Qian, Yi Yang, Yifei Hua, Chan Wang, Xiao Lin, Tong Cai, Dexin Ye, Erping Li, Ido Kaminer, and Hongsheng Chen has been reviewed.

The manuscript presents a new and elegant way to achieve superscattering without relying on embedded particles in extreme environment or relying on multiple resonances. Here the authors propose to utilize gain at the nanoparticle position to assist superscattering behavior. The difference

with existing approaches is that superscattering can occur with a single resonance by compensating the radiation leakage relying on spatial and temporal gain modulation. A reshaping of the radiation patterns in the superscattering regime is also observed.

I found the idea quite innovative and certainly worth publication.

I have a few comments:

The discussion on the steady state cross section affecting the transient response is quite paradoxal, at least to me. For device with extremely large cross section, I would expect extremely fast relaxation but instead the numerical results indicate the opposite. It might be worth elaborating on this unintuitive observation. Is it because of the radiation leakage compensation by amplification that artificially increases the mode lifetime? Reciprocally, the energy accumulation takes longer as well.

The manuscript reports about 40 times the scattering cross section in the active regime. It would be interesting to show how this number scales as a function of relevant parameter such as gain/loss ratio.

I found strange links in pdf, such as line 137, that directly point to Wikipedia...

Response Letter to Reviewers

We are grateful for the constructive comments on this manuscript (NCOMMS-22-12290) from all the referees. In the text below, each comment is quoted in *italics* and is followed by the corresponding detailed response. We have also revised the manuscript and supplementary material accordingly. These updates are highlighted in blue and by a vertical red line in the left margin in those files. In the text below, the references to these updates are highlighted in a similar way (i.e., by a vertical red line).

General comments from Referee #1:

In this manuscript, the authors proposed novel scattering enhanced method that overcomes the single-channel scattering limit by exploiting gain media. As a demonstration, the authors have design a two-dimensional resonator with tunnel diode in order to provide a negative differential resistance. Overall, the authors have presented an interesting concept to exploit gain metasurfaces.

The manuscript is well-written and structured. It shows studies on the dynamic of these media and it shows a progress in the scattering enhancement limit with respect previous achievements. However, some points must be addressed for publication in Nature Communications.

Authors Response:

We thank the referee for his/her positive comments. In the following, we address the specific comments point-by-point whilst revising our manuscript.

Specific comments from Referee #1:

Referee #1 -- Comment 1:

1-The authors claim that the total scattering cross section is more than 40 times the single-channel scattering at 3.55GHz, however no experimental point at the maximum enhancement is provided. Experimental point in this frequency must be provided in order to demonstrate the claimed enhancement.

Authors Response:

We thank the referee for pointing this out. As suggested, we have conducted a new experiment to demonstrate the scattering enhancement at 3.55 GHz. Figure R1 shows the near-field distribution and radar cross section C_{rcs} . From Fig. R1, the measured near-field distribution is consistent with the simulated case. To quantitatively characterize the scattering performance, we calculated the total scattering cross section C_{sct} via $C_{sct} = \frac{1}{2\pi} \int_0^{2\pi} C_{rcs} d\phi$. The calculation result turns out that C_{sct} is 41.9 times the single-channel scattering limit.

In the new submission, we have replaced the middle panel of Fig. 4d by Fig. R1 and added the description on lines 194-195 of the main text.

“Experimental result indicates that the total scattering cross section is more than 40 times the single-channel scattering limit at 3.55 GHz.”

Figure R1 | Experimental result of the single-channel superscatterer at 3.55 GHz. **a**, Near-field simulation and measurement of the total field in a region close to the single-channel superscatterer. **b**, Simulated and measured radar cross sections.

Referee #1 -- Comment 2:

2- An experimental measurement of the dielectric rod without gain metasurface could be very helpful to discard border effects in the electromagnetic set up measurements.

Authors Response:

This is an important question. In the new version, we have added the experimental evidence for the dielectric rod without gain metasurface in Fig. R2. Compared with the dielectric rod with gain metasurfaces (Fig. 4 in the main text and Fig. R1), the wavefront is flat and the scattering is almost negligible for pure dielectric rod. In addition, the C_{rCS} is very low (Fig. R2b), in a good agreement with the simulated case.

In the new submission, we have added the above discussion in the supplementary materials and referred to it on lines 237-238.

“Before the experiment, we measured the near-field for the pure dielectric rod without gain metasurfaces to check the border effects (Supplementary Fig. 12).”

Figure R2 | Experimental result of the pure dielectric rod without the gain metasurface. For conceptual demonstration, the measured frequency is 3.55 GHz. **a**, Near-field simulation and measurement of the total field in a region close to the dielectric rod. **b**, Simulated and measured radar cross sections. Notice that the scale in **b** is much smaller than that in Fig. R1.

Referee #1 -- Comment 3:

3-Taking into account that the experimental design and the results are based in the numerical simulation of CST studio commercial software, the numerical method must be more detailed (boundary conditions, mesh, how the metallic materials are considered, etc.)

Authors Response:

Thanks for the careful reading. In the new submission, we have added the simulation details in Methods section; see below.

“The scattering parameters of the gain metasurfaces are simulated using CST Microwave studio. The gain metasurface comprises copper patch and substrate layer with a relative permittivity of 3.4 and a thickness of 0.05 mm. A tunnel diode is incorporated into each metasurface element, whose equivalent circuits are embedded in the simulation. We applied the EM-circuit co-simulation to connect the metasurface and tunnel diode. The “FSS, Metamaterial-Unit Cell template” in the “Frequency domain solver” is chosen, with unit cell boundary conditions enabled in the x- and z-directions and open boundary conditions in the y-direction (Fig. 3a). Finally, an adaptively refined tetrahedral mesh was applied to the simulated structures with a number of about 35,000 cells. The maximum mesh size was set at approximately 4.8 mm, which is about one-eighteenth of a free-space wavelength at 3.5 GHz.”

General comments from Referee #2:

The manuscript entitled «Breaking the fundamental scattering limit with gain metasurfaces » by Chao Qian, Yi Yang, Yifei Hua, Chan Wang, Xiao Lin, Tong Cai, Dexin Ye, Erping Li, Ido Kaminer, and Hongsheng Chen has been reviewed.

The manuscript presents a new and elegant way to achieve superscattering without relying on embedded particles in extreme environment or relying on multiple resonances. Here the authors propose to utilize gain at the nanoparticle position to assist superscattering behavior. The difference with existing approaches is that superscattering can occur with a single resonance by compensating the radiation leakage relying on spatial and temporal gain modulation. A reshaping of the radiation patterns in the superscattering regime is also observed.

I found the idea quite innovative and certainly worth publication.

Authors Response:

We are grateful to the referee for these positive comments and acknowledging that “*I found the idea quite innovative and certainly worth publication*”. In the following, we address the specific comments point-by-point and revise our manuscript correspondingly.

Specific comments from Referee #2:

Referee #2 -- Comment 1:

I have a few comments:

The discussion on the steady state cross section affecting the transient response is quite paradoxal, at least to me. For device with extremely large cross section, I would expect extremely fast relaxation but instead the numerical results indicate the opposite. It might be worth elaborating on this unintuitive observation. Is it because of the radiation leakage compensation by amplification that artificially increases the mode lifetime? Reciprocally, the energy accumulation takes longer as well.

Authors Response:

We thank the referee for these very constructive suggestions. The numerical results are actually consistent with our expectation—a larger scattering cross section requires longer time for accumulating and releasing energy; this agrees with the study [*Phys. Rev. Lett.* 106, 165503 (2011)].

First, let's summarize the three situations we consider, i.e., superscatterer with gain, superscatterer without gain, and pure dielectric rod without gain. Their normalized scattering cross sections are 44.5, 4.2, and 0.46, respectively, corresponding to the buildup time of $\Delta t_1 = 28.5 \text{ ns}$, $\Delta t_2 = 4.1 \text{ ns}$, $\Delta t_3 = 0.8 \text{ ns}$; see Fig. R3.

According to the above results, we found that an object with larger scattering cross section corresponds to longer buildup time ($\Delta t_1 > \Delta t_2 > \Delta t_3$). This is understandable because the larger the total scattering cross section, the longer the time required to accumulate sufficient energy to reach a stable state. Reciprocally, the relaxation time is longer. In contrast, if an object has a tiny scattering cross section, the buildup/relaxation time will be very short, as exemplified in Fig. R3c.

For a gain system (Fig. R3a), when the incident wave cuts off abruptly, the field scattered by the single-channel superscatterer decays slowly. This exotic scattering phenomenon occurs not only because it has accumulated much scattering energy in the buildup period, but also the radiation leakage is compensated by the gain metasurfaces in the release period. This is in line with the referee's prediction that *"the radiation leakage compensation by amplification that artificially increases the mode lifetime. Reciprocally, the energy accumulation takes longer as well."*

Figure R3 | Transient response of the superscatterer with gain, superscatterer without gain, and pure dielectric rod. Their working frequency and physical size are the same. However, their scattering cross sections are different (noted in the figure), corresponding to different buildup time.

In the new submission, we have incorporated the above discussion on lines 131-135.

“The larger the total scattering cross section, the longer the time required to accumulate sufficient energy to reach a stable state. In a similar vein, the accumulated energy is stored for a longer period after the incident wave is cut off. For the gain system, the radiation leakage is compensated by the gain metasurfaces in the release process that artificially increases the mode lifetime.”

Referee #2 -- Comment 2:

The manuscript reports about 40 time the scattering cross section in the active regime. It would be interesting to show how this number scale as a function of relevant parameter such as gain/loss ratio.

Authors Response:

This is a good point. For our single-channel superscatterer, an ultrathin gain metasurface is used to offer a complex surface impedance Z_s . The real part of Z_s determines the lossy or gain property, and its imaginary part determines the capacitive or inductive reactance. Using the simulated annealing algorithm, we design a single-channel superscatterer with the normalized scattering cross section of 44.5 at 3.733 GHz; the specific parameters can be found in the main text.

Based on this superscatterer structure, we change the gain/loss state to check the scattering cross section (Fig. R4). In a lossy system with $Re(Z_s) > 0$, the scattering strength is inversely proportional to the loss. However, in a gain system with $Re(Z_s) < 0$, the scattering strength could increase remarkably. In principle, an infinitely large scattering cross section could be anticipated.

Figure R4 | Scattering cross section related to loss/gain. The five-pointed star represents the working point for our single-channel superscatterer. Z_s^0 represents the surface impedance provided by the gain metasurfaces in the single-channel superscatterer. $Re(Z_s^0) = -10.85 \Omega$ at the working frequency of 3.733 GHz. Here we vary $Re(Z_s)/Re(Z_s^0)$ to modify the loss/gain state.

In the new submission, we have incorporated the above discussion in the supplementary materials and noted it on lines 154-156.

“The increase/decrease of $Re(Z_s)$ (gain component) will affect the total scattering cross section; see Supplementary Fig. 5.”

Referee #2 -- Comment 3:

I found strange links in pdf, such as line 137, that directly point to Wikipedia ...

Authors Response:

Thanks for the careful reading. We have amended the strange links and checked the paper thoroughly.

REVIEWERS' COMMENTS

Reviewer #1 (Remarks to the Author):

The authors have answered to the review properly and extensively. I consider the final manuscript is improved with the previous version and I think must be considered for publishing in the journal.

Reviewer #2 (Remarks to the Author):

The comments of this manuscript have been carefully addressed and the manuscript should be considered for publication in Nat. Comm.

Response Letter to Reviewers

We are grateful for the constructive comments on this manuscript (NCOMMS-22-12290A) from all the referees. These comments are very valuable and helpful for improving our manuscript. In the text below, each comment is quoted in *italics* and is followed by the corresponding response.

Comments from Reviewer #1:

The authors have answered to the review properly and extensively. I consider the final manuscript is improved with the previous version and I think mus be considered for publishing in the journal.

Authors Response:

We thank the referee for the strong positive comments and for helping to improve our manuscript

Comments from Reviewer #2:

The comments of this manuscript have been carefully addressed and the manuscript should be considered for publication in Nat. Comm.

Authors Response:

We thank the referee for the positive comments and the recommendation of our work.